# Three-Dimensional Analytical Solution of Self-potential from Regularly Polarized Bodies in Layered Seafloor Model

Pengfei Zhang[1,2,3], Yi-an Cui[1,2,3], Jing Xie[1,2,3], Youjun Guo[1,2,3], Jianxin Liu[1,2,3], and Jieran Liu[1,2,3]

[1]School of Geosciences and Info-Physics, Central South University, Changsha, 410083, China
[2]Laboratory of Non-ferrous Resources and Geological Hazard Detection, Central South University, Changsha, 410083, China
[3]Key Laboratory of Metallogenic Prediction of Nonferrous Metals, Ministry of Education, Central South University, Changsha, 410083, China

**Correspondence:** Yi-an Cui (cuiyian@csu.edu.cn)

**Abstract.** The self-potential (SP) method is a sensitive geophysical technique to locate seafloor polymetallic sulfide deposits. A reasonable SP forward modeling can provide a good foundation for inversion and interpretation of the measured data. Based on the mirror image current theory, we proposed a method to derive the three-dimensional analytical solution of the SP generated by regularly polarized bodies in layered media, which is explained in detail within the context of the models. We discussed the analytical solutions for different types of layered models, considering variations in the number of layers and the distribution of sources. A lab-based oxidation-reduction experiment was conducted to record SP data. These data are used to simulate the SP generated by seafloor massive sulfide(SMS) deposits and validate the analytical solution previously. The result shows that the measured SP data matches the analytical solution well, demonstrating the correctness of the proposed method and the corresponding analytical solution. This approach is significant for achieving fast and precise forward modeling and inversion in SMS explorations.

## 1 Introduction

Seafloor Massive Sulfide(SMS) deposit is an important strategic resource for its rich gold, silver, copper, zinc, and other high-value metal ore (Mendonca, 2008). The research of submarine hydrothermal vents at the Galapagos in 1977 is the beginning of the seafloor massive sulfide which continues today (Corliss et al., 1979). More than 700 submarine hydrothermal anomalies have been identified, with over 100 regions currently recognized as having significant exploration potential (Hannington et al., 2011). The seafloor acts as a unique redox interface, where electrical conductors formed by mineral deposits generate electric currents as they traverse this boundary(Sato and Mooney, 1960; Jones, 1999). The self-potential method (SP) is a passive source method and needs no power source during nature conditions. The SP survey exhibits a distinct sensitivity to these anomalous electric currents, allowing for the rapid identification of SMS deposits. Corwin was the first to attempt to measure the SP signal in marine minerals with an offshore SP array and recorded an abnormal signal of up to 300 mV (Corwin, 1976). Safipour et al. recorded both horizontal components of a known site containing an SMS occurrence and proved that the SP method is an effective exploration tool in SMS areas with hydrothermal activity (Safipour et al., 2017). Kawada and Constable observed SP signals of SMS with a deep-tow handled an AUV(Autonomous Underwater Vehicle) respectively, which further

proved the SP method is useful in SMS exploration (Kawada and Kasaya, 2017; Constable et al., 2018). Zhu et al. reported a
deep-sea self-potential investigation at the Yuhuang hydrothermal field, where a horizontal array of electrodes detected negative
self-potential anomalies (~ -27 mV) and high electrical conductivities (up to 12 S/m), attributed to sulfide mineralization and
corrosion of polymetallic sulfides(Zhu et al., 2020). Su et al. used an autonomous underwater vehicle to take a SP survey on
the ultra slow-spreading Southwest Indian Ridge with a water depth from 1300m to 2200m. And a 3D SP tomography was
used to reveal an ore-body with a vertical extent of 100m (Su et al., 2022). The above research suggest that the self-potential
method contributes to seafloor massive sulfide surveys.

In addition to field studies, laboratory experiments have been instrumental in advancing our understanding of self-potential
phenomena. Castermant explored how redox potential distributions are inferred from self-potential measurements during the
corrosion of buried metallic bodies in a controlled sandbox experiment(Castermant et al., 2008). Martínez-Pagán investi-
gated the use of self-potential monitoring to detect and track the leakage and migration of a salt plume in a sandbox experi-
ment(Martínez-Pagán et al., 2010). Fachin presented a laboratory experiment exploring SP signals generated by a biogeobattery
model, simulating electron transfer between organic matter and oxygen-rich sediments(Fachin et al., 2012). Vasconcelos exam-
ined the relationship between self-potential signals and streaming potentials generated by water flow in porous media through
laboratory experiments(Vasconcelos et al., 2014). Given the complexity of layered seafloor environments, validating analytical
solutions through controlled laboratory experiments becomes crucial. By creating a controlled, layered model structure in the
laboratory, we can systematically test and validate our analytical solution under known conditions. This approach allows us to
bridge the gap between theoretical modeling and measured SP data, ensuring the reliability and applicability of our method to
more complex seafloor scenarios.

The forward modeling process simulates the interaction between current sources and ore bodies, predicting the self-potential
distribution. This enables more precise interpretation of observed data and improves the inversion of subsurface mineral de-
posits' geometry and electrical properties(Minsley, 1997). The forward methods commonly used for self-potential methods
include numerical solutions and analytical solutions(Xie et al., 2023; Minsley et al., 2007). The numerical solution is a qualita-
tive (or semi-quantitative) technique (Wei et al., 2023), which includes the finite element method (Alarouj and Jackson, 2022;
Bérubé, 2007), the finite volume method (Sheffer and Oldenburg, 2007), the finite difference method (Mendonca, 2008), the
natural-infinite element coupling method (Xie et al., 2020a), the finite-infinite element coupling method (Xie et al., 2020b)and
so on. Numerical modeling applies to any complex model. Xie et al. proposed a finite-infinite element coupling method to
calculate a numerical model of the marine SP from seafloor hydrothermal sulfide deposits (Xie et al., 2021). However, the
result of numerical method is obtained by approximate calculation under a certain condition(Chandra et al., 2020). The con-
ductivity structure of complex media will affect the composition of the stiffness matrix. For anomalous sources that are not
uniformly polarized, their uncertainty will also impact the construction of the source term in the finite element system equa-
tions. And the complex artificial boundary conditions also limit its development. Compared to numerical methods, analytical
solutions are strict formulas that can overcome the difficulties in solving the Poisson equation. In most studies, the polarization
structure of ore bodies can be equivalent to special geometry shapes(Yungul, 1950; Ai et al., 2024). The analytical solution
of polarized geometry body is significant in mineral exploration(Luo et al., 2023; Liu et al., 2023). Yungul discussed the an-

alytical solution of a polarized sphere and other researchers get the analytical solution of SP anomaly along a profile passing
over the centre of the sphere or to the strike of a horizontal cylinder (Yungul, 1950; Bhattacharya and Roy, 1981; El-Araby,
2004). Murthy and Haricharan discussed the analytical solution of SP anomaly at any point on a profile perpendicular to the
strike of a 2-D inclined thin sheet (Satyanarayana Murty and Haricharan, 1985). Further, Biswas derived the expression of SP
anomaly analytical solution when the sheet parameters were described with respect to one edge of the sheet and in terms of
the $X$ and $Z$ coordinate of the top and bottom edge of the sheet(Biswas and Sharma, 2014). Dmitriev derived the analytical
solution of SP anomaly due to a thick dipping body which could represent an ore body(Dmitriev, 2012). In marine fieldwork,
it's a challenge to determine the center or strike of subsurface geometric bodies accurately. The survey lines typically do not
pass directly over the anomaly. Two-dimensional analytical solutions, while useful for simplified scenarios, may fall short in
large-scale forward and inverse modeling. The inferred location and polarization angle of an anomalous source based on 2D
solutions may not correspond to the true source properties. Furthermore, existing analytical solutions are predominantly based
on homogeneous half-space conditions, assuming a uniform subsurface medium. Unlike terrestrial environments, most current
marine self-potential measurement systems struggle to achieve full ground contact(Safipour et al., 2017; Kawada and Kasaya,
2017, 2018; Constable et al., 2018). When the measurement system is positioned within seawater, deriving analytical solutions
for layered media becomes crucial. To address this, we proposed a 3D analytical solution based on the mirror image method
for layered SMS models. The analytical solution serves as benchmarks for numerical simulations, enabling us to identify and
correct deviations in numerical approaches. Moreover, in scenarios where the analytical model is applicable, it offers faster
computations compared to numerical methods. This not only enhances computational efficiency but also provides a foundation
for the inversion and interpretation of measured data.

## 2    The mirror image method of electric dipole

The mirror image method is based on the uniqueness theorem. It can be used to solve the electrostatic field problem such as
some special problems of conductor boundary with point source or line source(Stephenson, 1990). By introducing a virtual
image dipole on the other side of the medium boundary, the boundary conditions for the electric field and potential are satisfied.
This allows the originally complex multilayer medium problem to be treated as a problem in a uniform half-space medium.
The uniqueness theorem states that there is only one solution in the electrostatic system when the boundary conditions are
uniquely determined(Wang et al., 2019). A seafloor massive sulfide model which meets the uniqueness theorem is built as
shown in Fig.1. *XOY* surface is the boundary between the sea and the air. We suppose the depth of seawater is D and the depth
of the seafloor is L. A three-dimensional coordinate system is established with vertical sea level downward as the Z-axis. We
use $\varepsilon$, $\mu$, $\sigma$ to denote the medium permittivity, magnetic conductivity and conductivity and use subscripts 0,1,2 to denote air,
seawater and seafloor. We assume there is an electric dipole $P = Idl$ oriented in any direction at the $(x_0, y_0, z_0, z_0<D)$, and the
measuring point is at $(x, y, z)$. If $z \leq 0$, the measuring point is in the air or on the sea surface. If $0 <z< D$, the measuring point
is in seawater. We decompose the electric dipole, oriented in any direction, into a horizontal dipole parallel to the Z=0 plane

and a vertical dipole parallel to the Z-axis. The following derivation process is based on the example of a horizontal electric dipole $P_x = I_x dl$.

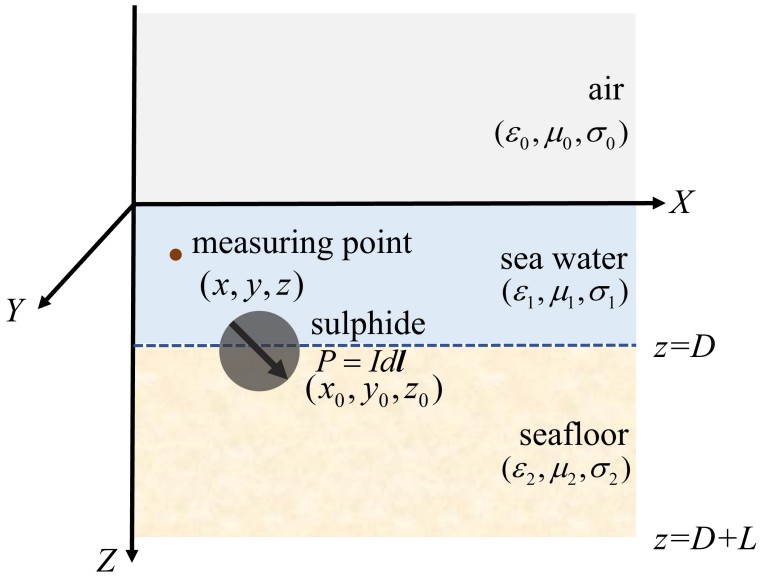

**Figure 1.** Sketch of SMS simplified model. The model includes air, seawater and seafloor. The Z axis points down towards the seafloor. The black sphere represents the simplified sulfide ore body, and the black arrow indicates its polarization direction.

## 2.1 Potential equivalence of the sphere and the electric dipole

An uneven double electric layer forms on the surface of the polarized sphere. The potential difference $\Delta\varepsilon$ varies linearly with
the direction of polarization, which can be expressed as

$$\Delta\varepsilon = \Delta U_0 \cos\theta \tag{1}$$

where $\Delta U_0$ is the maximum potential difference. $\theta$ is the angle between the polarization axis and the line from the measuring point to the sphere center. In a uniformly polarized sphere, the external potential $U$ is distributed symmetrically about the polarization axis and is independent of the azimuthal angle. This specific symmetry leads to a simplified form of the Laplace
equation :

$$\frac{\partial}{\partial R}(R^2 \frac{\partial U}{\partial r}) + \frac{1}{\sin\theta} \cdot \frac{\partial}{\partial\theta}(\sin\theta \frac{\partial U}{\partial\theta}) = 0 \tag{2}$$

The general solution of potential can be solved by separation of variables as

$$U = \sum_{n=0}^{\infty}(A_n R^n + B_n/R^{n+1})P_n(\cos\theta) \tag{3}$$

where $P_n(cos\theta)$ is Legendre polynomial of n, $An$ and $Bn$ is undetermined coefficient. As $R \to \infty$, the potential outside the sphere $U_1 \to 0$. As $R \to 0$, the potential inside the sphere $U_2 \to 0$. The potentials inside ($U_2$) and outside ($U_1$) the sphere can be expressed as:

$$U_1 = \sum_{n=0}^{\infty} \left( \frac{B_n}{R^{n+1}} \right) P_n(\cos\theta) \tag{4}$$

$$U_2 = \sum_{n=0}^{\infty} \left( A_n R^n \right) P_n(\cos\theta) \tag{5}$$

Based on the boundary conditions:

(1)There is a potential jump on both sides of the sphere. When $R=r_0$, we have:

$$\Delta\varepsilon = U_2 - U_1 = \Delta U_0 \cos\theta \tag{6}$$

where $U_1$ and $U_2$ is the potential outside and inside the sphere.

(2)The current density normal vectors are continuous on both sides of the sphere. When $R=r_0$ we have:

$$\frac{1}{\rho_1} \frac{\partial U_1}{\partial R} = \frac{1}{\rho_2} \frac{\partial U_2}{\partial R} \tag{7}$$

It can be obtained that:

$$A_1 = -\frac{2\rho_2}{2\rho_2 + \rho_1} \cdot \frac{\Delta U_0}{r_0}$$
$$B_1 = \frac{\rho_1}{2\rho_2 + \rho_1} \cdot \Delta U_0 \cdot r_0^2 \tag{8}$$

For the self-potential generated by a simplified polarized body in a uniform half-space, we can directly handle the interface effects by doubling(Li et al., 2005; Biswas, 2021). From the formula (5), we obtain:

$$U = 2 \cdot U_1 = \frac{2\rho_1}{2\rho_2 + \rho_1} \cdot \frac{r_0^2}{R^2} \cdot \Delta U_0 \cos\theta = M \cdot \frac{\cos\theta}{R^2}$$
$$M = \frac{2\rho_1}{2\rho_2 + \rho_1} r_0^2 \Delta U_0 \tag{9}$$

where $\theta$ is the polarization angle, $R$ is the distance between the measuring point and the center of the sphere, $\rho_1$ is the resistivity of the medium, $\rho_2$ is the resistivity of the sphere and $r_0$ is the radius of the sphere. The scalar potential caused by a constant electric dipole is given by the formula

$$U = \frac{Idl}{4\pi\sigma} \cdot \frac{(-x)}{R^3} = -P_0 \frac{x}{R^3} \tag{10}$$

where $\frac{x}{R} = cos\theta$. In equation (9) and (10), $R=r$, $P_0=M$. And we get the potential distribution along the surface of a uniformly polarized sphere is equivalent to an electric dipole.

## 2.2 Two layers of medium

When there is a two-layer medium model, we discuss the air-seawater model and the seawater-seafloor model. In the first model, we suppose the location of the image of the source $P'_x = I'_x dl$ is $(x_0, y_0, -z_0)$, when the measuring point is in the sea($z > 0$). It's assumed that the whole space is filled with seawater. We have the scalar potential of the source and the image:

$$U_{\text{sea}} = U_x + U'_x = \frac{I_x dl(x - x_0)}{4\pi\sigma_1 R_1^3} + \frac{I'_x dl(x - x_0)}{4\pi\sigma_1 R_0^3} (z > 0) \tag{11}$$

where $R_0 = (x - x_0)\mathbf{i} + (y - y_0)\mathbf{j} + (z + z_0)\mathbf{k}$, $R_1 = (x - x_0)\mathbf{i} + (y - y_0)\mathbf{j} + (z - z_0)\mathbf{k}$

If the measuring point is in the air($z \leq 0$), the boundary condition requires that the potential in the air matches the potential just below the interface in the seawater. The key boundary conditions that need to be satisfied at the air-sea interface include the continuity of the electric potential across the interface and the continuity of the normal component of the electric field (or current density) across the interface. So the location of the image of the source is $(x_0, y_0, z_0)$, which is coincided with the source. We suppose the whole space is filled with air. The combined dipole moment $P''_x = I''_x dl$ is

$$U_{air} = \frac{I''_x dl(x - x_0)}{4\pi\varepsilon_0 R_1^3} (z \leq 0) \tag{12}$$

where $R_1 = (x - x_0)\mathbf{i} + (y - y_0)\mathbf{j} + (z - z_0)\mathbf{k}$.

It can be obtained from the boundary conditions of the mirror image theory:

(1) The potential of both sides of the surface is continuous($(U_{sea}|_{z\to 0+} = U_{air}|_{z\to 0-})$). We have:

$$\frac{I''_x}{\varepsilon_0} = \frac{I'_x + I_x}{\sigma_1} \tag{13}$$

(2) The current normal vectors on both sides of the interface are continuous and satisfy the boundary condition $j_{1z}|_{z\to 0+} = j_{0z}|_{z\to 0-}$. We have $\sigma_1 \frac{\partial U_{sea}}{\partial z}\big|_{z\to 0+} = \sigma_0 \frac{\partial U_{air}}{\partial z}\big|_{z\to 0-}$. Because in the air $\sigma_0 = 0$, only $\sigma_1 \frac{\partial U_{sea}}{\partial z}\big|_{z\to 0+} = 0$ can satisfy the boundary condition, we have:

$$I'_x = I_x \tag{14}$$

Using equations (12) and (13), we obtain

$$I''_x = \frac{2\varepsilon_0}{\sigma_1} I_x \tag{15}$$

The above analysis shows the horizontal dipole has two situations when it is in the air-seawater model. If the measuring point is in the sea, the location of the mirror image $I'_x dl$ is $(x_0, y_0, -z_0)$. If the measuring point is in the air, the mirror image coincides with the source $(x_0, y_0, z_0)$ and the combined dipole is $\frac{2\varepsilon_0}{\sigma_1} I_x dl$. In the seawater-seafloor model, we can calculate like the first model. We suppose the electric dipole source $I_x dl$ is at $(x_0, y_0, z_0)$. If we measure in the seawater, we can get the mirror image $\frac{\sigma_1 - \sigma_2}{\sigma_1 + \sigma_2} I_x dl$ is at $(x_0, y_0, -z_0)$. If the measuring point is on the seafloor, the mirror image is at $(x_0, y_0, z_0)$ which the combined dipole is equivalent to $\frac{2\sigma_2}{\sigma_1 + \sigma_2} I_x dl$. To verify the correctness of the mirror image method, we compare the 2D analytical solution

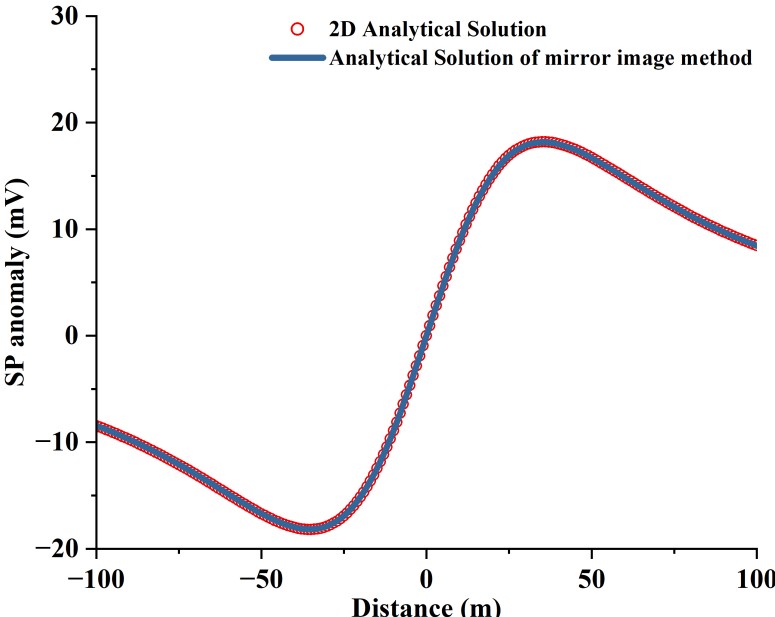

**Figure 2.** The comparison of the analytical solution of the seawater-seafloor model. The analytical solution of the mirror image method is consistent with the 2D analytical solution.

in homogeneous half-space and the analytical solution of the seawater-seafloor model when the measuring lines fully contact seafloor. Based on the derivation above, the potential anomaly measured in the seawater of sea water-seafloor model can be expressed as:

$$U = \frac{I x dl (x - x_0)}{4\pi\sigma_1 [(x - x_0)^2 + (-z + z_0)^2]^{3/2}} + \frac{\frac{\sigma_1 - \sigma_2}{\sigma_1 + \sigma_2} I x dl (x - x_0)}{4\pi\sigma_1 [(x - x_0)^2 + (z - z_0)^2]^{3/2}} \tag{16}$$

The 2D analytical solution in homogeneous half-space can be expressed as(Xie et al., 2021):

$$\phi = M \cdot \frac{x \cos a - h_0 \sin a}{(h_0^2 + x^2)^{3/2}} \tag{17}$$

where $M$ is the electric dipole moment, $\alpha$ is the polarizing angle and $h_0$ is the depth of the electric dipole.

When $\sigma_2 = \sigma_1$, the electrical conductivity of the two media (seawater and air, or seawater and seafloor) becomes equal. This effectively means that there is no boundary between the two media, and the system behaves as a single, uniform medium. The comparison results is shown in Fig.2. The two solutions appear to coincide closely, indicating that the mirror image method

accurately meet the traditional analytical solution. It proves the mirror image method is correct in calculating the polarization self-potential.

## 2.3 Three layers of medium

The actual ocean environment can be reduced to a three-layer model consisting of air, seawater and seafloor. The source "generates" countless mirror images among the three mediums. The source point generates corresponding images in the other two media. The generated images in turn create new images in the other medium. For instance, an image dipole generated in the air by the source point will produce a second image dipole in the seafloor medium; similarly, an image dipole generated in the seafloor medium will produce another second image dipole in the air. This process continues, generating an infinite number of image dipoles. In the ocean model shown in Figure 3, the potential produced by an electric dipole in the seafloor can be equivalent to the superposition of the source and an infinite number of mirror images. The potential generated by each image point in a manner similar to the two-layer model, based on the same boundary conditions. Upon solving for different image points, we divide mirror images into four categories for their different locations and dipole moments. The locations and potentials of these mirror images are shown in the table 1. The coordinates of the first type of image dipole are $(x_0, y_0, 2mD - z_0, m=1,2...)$, with the corresponding dipole moment solved as $\left(\frac{\sigma_1 - \sigma_2}{\sigma_1 + \sigma_2}\right)^m I_x d\mathbf{l} = \eta^m I_x dli$ and the position vector as $r_{1m} = (x - x_0)\mathbf{i} + (y - y_0)\mathbf{j} + (z - 2mD + z_0)\mathbf{k}$; The coordinates of the second type of image dipole are $(x_0, y_0, 2mD + z_0, m=1,2...)$, with the corresponding dipole moment solved as $\left(\frac{\sigma_1 - \sigma_2}{\sigma_1 + \sigma_2}\right)^m I_x dl = \eta^m I_x dli$ and the position vector as $r_{1m} = (x - x_0)\mathbf{i} + (y - y_0)\mathbf{j} + (z - 2mD + z_0)\mathbf{k}$; The coordinates of the third type of image dipole are $(x_0, y_0, -2nD + z_0, n=1,2...)$, with the corresponding dipole moment solved as $\left(\frac{\sigma_1 - \sigma_2}{\sigma_1 + \sigma_2}\right)^n I_x dl = \eta^n I_x dli$ and the position vector as $r_{1n} = (x - x_0)\mathbf{i} + (y - y_0)\mathbf{j} + (z + 2nD - z_0)\mathbf{k}$; The coordinates of the fourth type of image dipole are $(x_0, y_0, -2nD - z_0, n=1,2...)$, with the corresponding dipole moment solved as $\left(\frac{\sigma_1 - \sigma_2}{\sigma_1 + \sigma_2}\right)^n I_x dl = \eta^n I_x dli$ and the position vector $r_{2n} = (x - x_0)\mathbf{i} + (y - y_0)\mathbf{j} + (z + 2nD + z_0)\mathbf{k}$. The source dipole is included in the third type of image dipole(n=0). These four different types of image dipoles do not have physical differences; rather, they are classified based on their mathematical similarity observed during the actual solution process.

**Table 1.** Locations and dipole moments of the source and mirror images

| | location | Dipole moments | The position vector between the measuring point and the source |
|---|---|---|---|
| 1 | $(x_0, y_0, 2mD - z_0)$ | $\left(\frac{\sigma_1 - \sigma_2}{\sigma_1 + \sigma_2}\right)^m I_x d\mathbf{l} = \eta^m I_x dli\,(m=1,2,...)$ | $r_{1m} = (x - x_0)\mathbf{i} + (y - y_0)\mathbf{j} + (z - 2mD + z_0)\mathbf{k}$ |
| 2 | $(x_0, y_0, 2mD + z_0)$ | $\left(\frac{\sigma_1 - \sigma_2}{\sigma_1 + \sigma_2}\right)^m I_x dl = \eta^m I_x dli\,(m=1,2,...)$ | $r_{2m} = (x - x_0)\mathbf{i} + (y - y_0)\mathbf{j} + (z - 2mD - z_0)\mathbf{k}$ |
| 3 | $(x_0, y_0, -2nD + z_0)$ | $\left(\frac{\sigma_1 - \sigma_2}{\sigma_1 + \sigma_2}\right)^n I_x dl = \eta^n I_x dli\,(n=0,1,...)$ | $r_{1n} = (x - x_0)\mathbf{i} + (y - y_0)\mathbf{j} + (z + 2nD - z_0)\mathbf{k}$ |
| 4 | $(x_0, y_0, -2nD - z_0)$ | $\left(\frac{\sigma_1 - \sigma_2}{\sigma_1 + \sigma_2}\right)^n I_x dl = \eta^n I_x dli\,(n=0,1,...)$ | $r_{2n} = (x - x_0)\mathbf{i} + (y - y_0)\mathbf{j} + (z + 2nD + z_0)\mathbf{k}$ |

The scalar potential of the horizontal electric dipole $P_x = I_x dli$ at the measuring point for any dipole moment in the seafloor can be expressed as

$$\Phi_x(x,y,z) = \sum_{m=1}^{\infty}\left[\frac{\eta^m I_x dl(x - x_0)}{4\pi\sigma_1 r_{1m}^3} + \frac{\eta^m I_x dl(x - x_0)}{4\pi\sigma_1 r_{2m}^3}\right] + \sum_{n=0}^{\infty}\left[\frac{\eta^n I_x dl(x - x_0)}{4\pi\sigma_1 r_{1n}^3} + \frac{\eta^n I_x dl(x - x_0)}{4\pi\sigma_1 r_{2n}^3}\right] \tag{18}$$

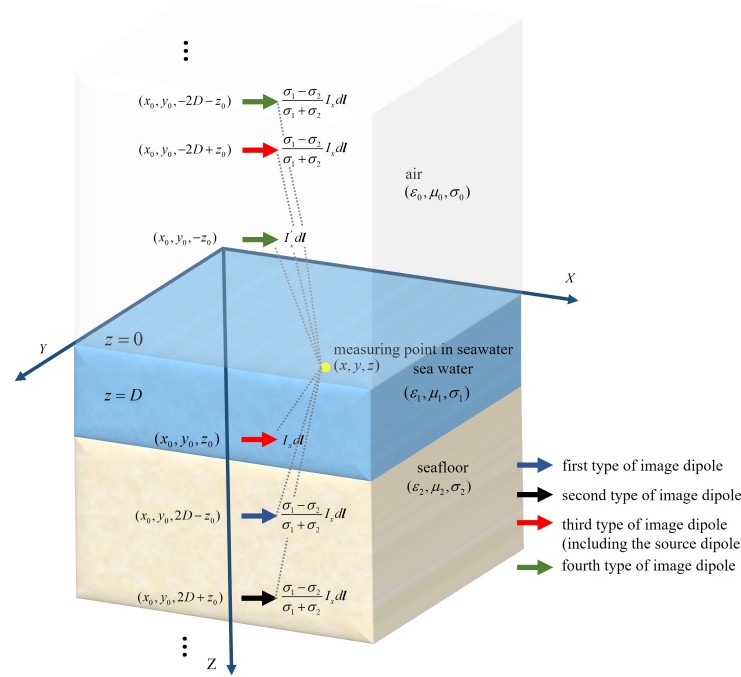

**Figure 3.** Sketch of three layers of medium in SMS model. The yellow point represents the measuring point, the arrows in different colors represent different kinds of electric dipoles .

The expression of $r_{1m}$, $r_{2m}$, $r_{1n}$, and $r_{2n}$ is shown in table 1. The electric dipoles in the other two directions $P_y=I_y dlj$ and $P_z=I_z dlk$ can be expressed by the same method. So we can get the potential of the electric dipole $P=I_x dli + I_y dlj + I_z dlk$ in any direction which can be equivalent to the superposition of the source and the countless mirror images:

$$U(x,y,z)=U_x(x,y,z)+U_y(x,y,z)+U_z(x,\text{y},z) =$$

$$\sum_{m=1}^{\infty}\left[\begin{array}{c}\dfrac{\eta^m I_x dl(x-x_0)}{4\pi\sigma_1 r_{1m}^3}+\dfrac{\eta^m I_y dl(y-y_0)}{4\pi\sigma_1 r_{1m}^3}-\dfrac{\eta^m I_z dl(z-2mD+z_0)}{4\pi\sigma_1 r_{1m}^3}\\[2mm]+\dfrac{\eta^m I_x dl(x-x_0)}{4\pi\sigma_1 r_{2m}^3}+\dfrac{\eta^m I_y dl(x-x_0)}{4\pi\sigma_1 r_{2m}^3}+\dfrac{\eta^m I_z dl(z-2mD-z_0)}{4\pi\sigma_1 r_{2m}^3}\end{array}\right] +$$

$$\sum_{n=0}^{\infty}\left[\begin{array}{c}\dfrac{\eta^m I_x dl(x-x_0)}{4\pi\sigma_1 r_{1n}^3}+\dfrac{\eta^n I_y dl(y-y_0)}{4\pi\sigma_1 r_{1n}^3}+\dfrac{\eta^n I_z dl(z+2mD-z_0)}{4\pi\sigma_1 r_{1n}^3}\\[2mm]+\dfrac{\eta^n I_x dl(x-x_0)}{4\pi\sigma_1 r_{2n}^3}+\dfrac{\eta^n I_y dl(y-y_0)}{4\pi\sigma_1 r_{2n}^3}-\dfrac{\eta^n I_z dl(z+2mD+z_0)}{4\pi\sigma_1 r_{2n}^3}\end{array}\right] \quad (19)$$

## 3   Analytical calculation of electric dipole potential distribution in SMS

In this chapter, we perform numerical simulations of dipoles in different orientations based on the image method. Additionally, we plot slices of the self-potential signals at various depths to investigate the impact of measurement depth on the observed

potential signals. We suppose the seawater depth is 100 m and the seafloor extends indefinitely along the Z axis. The electric dipole simplified by spherical SMS is at the seafloor surface with location (0,0,-100). The conductivity of the seawater ($\sigma_1$) and the seafloor ($\sigma_2$) is 4 S/m and 0.4 S/m. The dipole moments of both the horizontal and vertical dipoles are $3*10^5$ C·m. An inclined dipole can be decomposed into an X-direction horizontal dipole ($3*10^5$ C·m), a Y-direction horizontal dipole ($3*10^5$ C·m), and a vertical dipole ($3*10^4$ C·m). In the infinite summation, the computation will terminate when the difference between consecutive terms is less than $10^{-10}$. The three-dimensional potential distribution diagrams presented in Figure 4 illustrate the self-potential fields generated by different orientations of electric dipoles within a medium. Each subfigure provides a comprehensive visualization of the potential distribution and includes specific slices at various depths (z-coordinates) to offer detailed insights into the spatial variations of the potential. The horizontal electric dipole produces a positive SP anomaly and a negative SP anomaly on either side of it. The absolute values of the exceptions are equal. There is a negative anomaly caused by a vertical electric dipole. The tilted electric dipole produces a positive and a negative SP anomaly like the horizontal electric dipole. The self-potential signals increase along the polarization angle of the dipole, reflecting the combined effects of the horizontal and vertical components of the dipole's orientation. From the depth analysis, it is evident that at z=-99 meters, the potential distribution displays a distinct dipole field. The horizontal electric dipole field symmetrically extends along the horizontal axis. The vertical electric dipole field is concentrated around the dipole axis. The inclined electric dipole field is oriented along the dipole's tilt direction. At z=-80 meters, the potential distribution shows a more diffused pattern, and the potential is generally discernible. At z=-60 meters, the potential field further attenuates. Due to the complexity of mineralization, actual seafloor polymetallic sulfide deposits often present as multi-source polarized bodies. To study the characteristics of self-potential signals from multi-source polarized bodies, we perform forward modeling on a multi-source polarized body model. Under the same conditions, we assume (a) anomaly source 1 is located at (-25, 0, -100) and anomaly source 2 is located at (25, 0, -100), both being horizontally polarized; (b) anomaly source 1 is located at (-25, 0, -100) and is horizontally polarized, while anomaly source 2 is located at (25, 0, -100) and is inclined polarized (with the same polarization angle as previously described). The forward modeling results are shown in the Figure 5. The slices demonstrate that the inclined polarization of results (b) in an elongated and distorted SP pattern compared to the symmetrical pattern observed in scenario (a). Overall, the self-potential signals generated by the multi-source model are more complex, making it difficult to discern the trends of subsurface anomaly sources from the self-potential signals.

## 4  A experimental verification about 3D analytical solution of mirror image method

We built a three dimensional system for self-potential measurements from a laboratory perspective (shown in Fig.7) to prove the analytical solution. A tank, with a scale of 50cm×50cm×100cm, filled with sand and saline water to simulate the ocean environment. A 10 cm thick layer of quartz sand (average grain size 0.4 mm, porosity 0.51) was laid at the bottom of the tank. Ag-AgCl electrodes were used due to their lower noise and more stable measurements compared to other non-polarizing electrodes(Rowan et al., 2023). A system of 120 Ag-AgCl non-polarizing electrodes, each with a diameter of 6 mm, was embedded in a 3D-printed measurement device. The electrodes were arranged in a 24×5 grid, with a lateral spacing of 4 cm

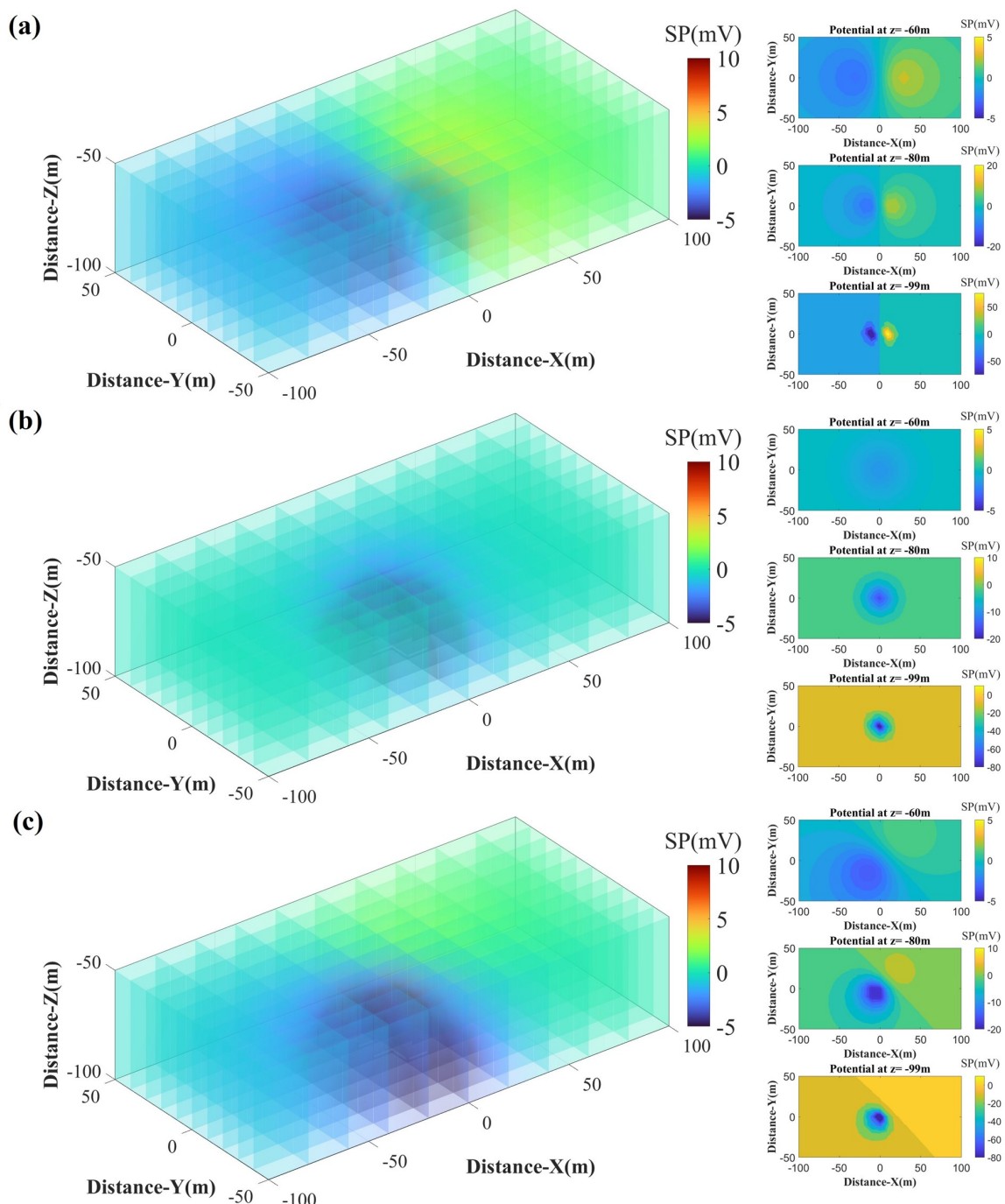

**Figure 4.** Three-dimensional potential distribution diagrams for different orientations of electric dipoles. (a) a horizontal electric dipole and its potential slices at z=-60, z=-80, and z=-99 meters. (b) a vertical electric dipole and its potential slices at z=-60, z=-80, and z=-99 meters. (c) an inclined electric dipole, which is tilted 45 degrees horizontally and 45 degrees vertically and its potential slices at z=-60, z=-80, and z=-99 meters.

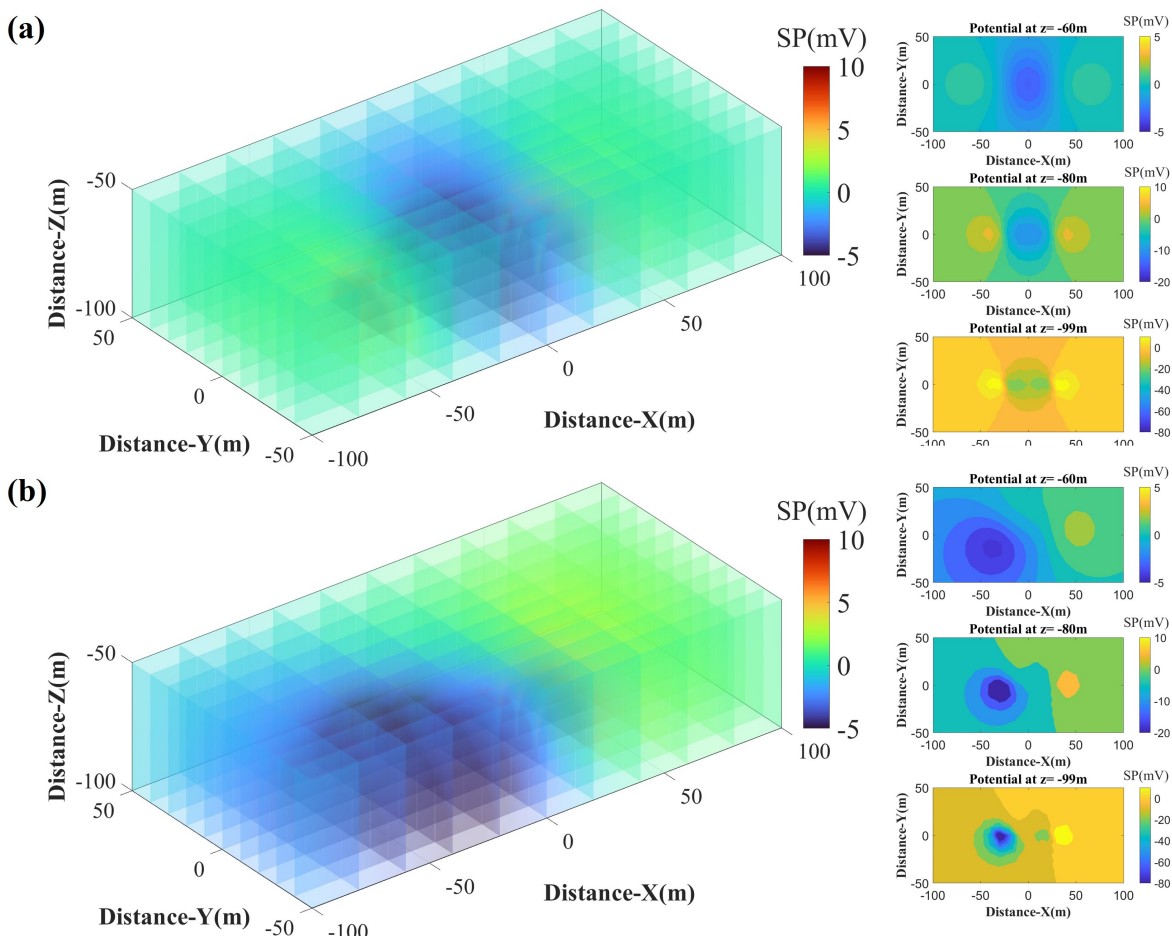

**Figure 5.** Three-dimensional potential distribution diagrams for multi electric dipoles. (a)two horizontally polarized electric dipole. (b) a horizontally polarized electric dipole and an inclined polarized electric dipole

and a longitudinal spacing of 6 cm. The measurement instrument used for the experiment was a multi-channel self-potential monitor with a sensitivity of 0.01 mV. To fully saturate the quartz sand at the bottom, saline water (maintaining the same salinity as seawater at 35‰) was injected through the bottom inlet of the tank, reaching a depth of 15 cm. After allowing the tank to settle for 4-5 days, the suspended sand in the water settled. The water level was recorded every two days, and saline water was added to prevent changes in salinity due to evaporation. The room temperature was maintained at $26 \pm 2°C$ throughout the experiment.

A sphere made of copper and iron was placed between the sand and the saline water. When a copper-iron sphere is immersed in water, the iron acts as the anode and undergoes oxidation, releasing electrons. Oxygen acts as the cathode and undergoes reduction, consuming electrons. Copper does not participate in the chemical reactions but serves as a medium for electron transport. Electrons within the sphere transfer from the iron hemisphere to the copper hemisphere, while electrons on the

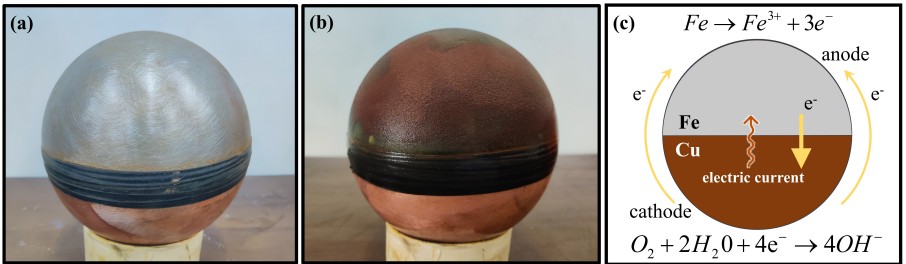

**Figure 6.** (a) A sphere composed of copper and iron, with the black part being insulating tape. (b)The sphere after the experiment shows significant oxidation and rust formation on the iron at the upper part of the sphere. The redox process of the sphere and its electrochemical half-cell reactions.

exterior surface migrate from the copper hemisphere to the iron hemisphere. This process ultimately results in the formation of a self-potential. The redox reactions occurred on the surface of the sphere with the electronic transfer. So we can control the polarization orientation of the electric dipole by changing the polarization angle of the sphere. We measured the SP signal when the Cu-Fe interface and the *XOY* plane were at an angle of 0° and 45° to simulate a vertical and an inclined electric dipole. The spheres before and after the experiment are shown in Figure 6. Between the two sets of experiments, we thoroughly washed and polished the rusted sphere until all rust was removed. The measurement results and forward modeling results are shown in Figure 8. The white dashed circle marks the projection position of the sphere. The red line represents the best-fit linear regression, indicating the correlation between the experimental data and the model predictions. It can be seen that the self-potential signal characteristics of the experimental results and the forward modeling simulations are generally consistent. The inclined sphere clearly shows potential characteristics corresponding to the polarization direction of the sphere, while the vertically placed sphere exhibits a significant negative potential characteristic at the sphere's location. We present the differences between the forward modeling parameters and the experimental parameters in Table 2. ME and Std represent the mean error and the standard deviation of error. Since the dipole moment of the sphere is an estimated value, the coefficient of determination ($R^2$) is 0.68 for the vertical sphere and 0.67 for the inclined sphere, ,which nonetheless reflects a good correlation between the experimental data and the model predictions. The expression for $R^2$ is as follows:

$$R^2 = 1 - \frac{\sum_{i=1}^{n}(y_i - \hat{y}_i)^2}{\sum_{i=1}^{n}(y_i - \bar{y})^2} \tag{20}$$

where $y_i$ is the observed value, $\hat{y}_i$ is the model predicted value, $\bar{y}$ is the mean of the observed values, and $n$ is the number of samples.

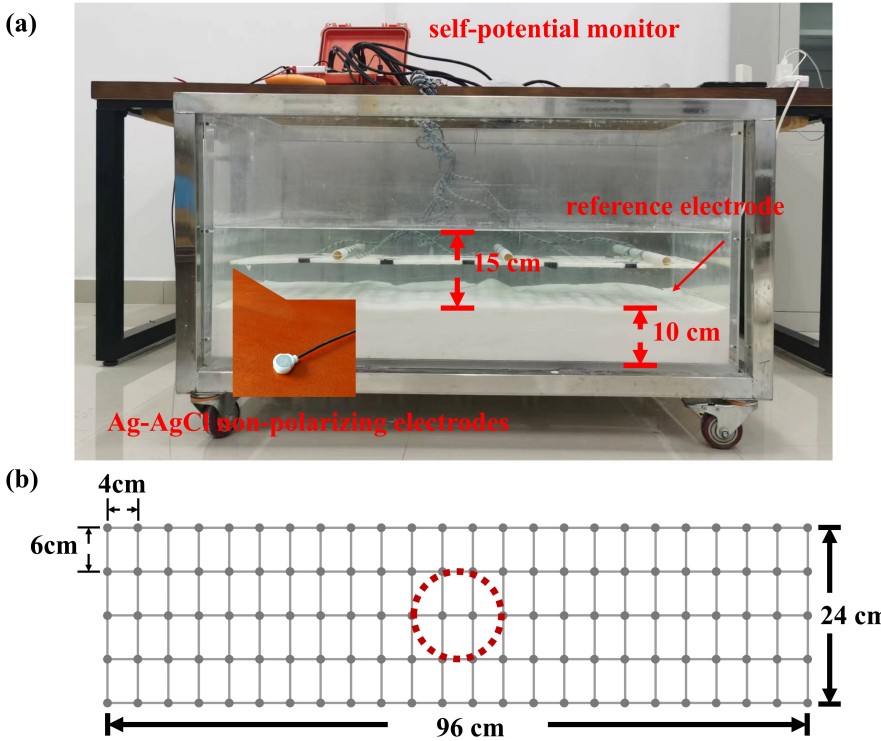

**Figure 7.** Sketch of SP measuring system. A Fe-Cu sphere was placed in the interface of saline water and the sand. According to the time-lapse data, the redox process was stable after 20 hours. We used the stabilized polarization data for analysis.

**Table 2.** Comparison between numerical simulation and experimental results

|  | *Inclined polarized sphere* | *Forward modeling of the inclined dipole* | *Vertical polarized sphere* | *Forward modeling of the vertical dipole* |
|---|---|---|---|---|
| $x_0$ | 48 | 48 | 48 | 48 |
| $y_0$ | 12 | 12 | 12 | 12 |
| $z_0$ | 16 | 16 | 16 | 16 |
| $I_x dl$ | - | 0 | - | 0 |
| $I_y dl$ | - | 0 | - | 664 |
| $I_z dl$ | - | -759 | - | 660 |
| n | - | 15 | - | 15 |
| $\sigma_2$ | - | $10^6$ | - | $10^6$ |
| ME | 0.049 | | 1.309 | |
| Std | 2.62 | | 2.86 | |
| $R^2$ | 0.68 | | 0.67 | |

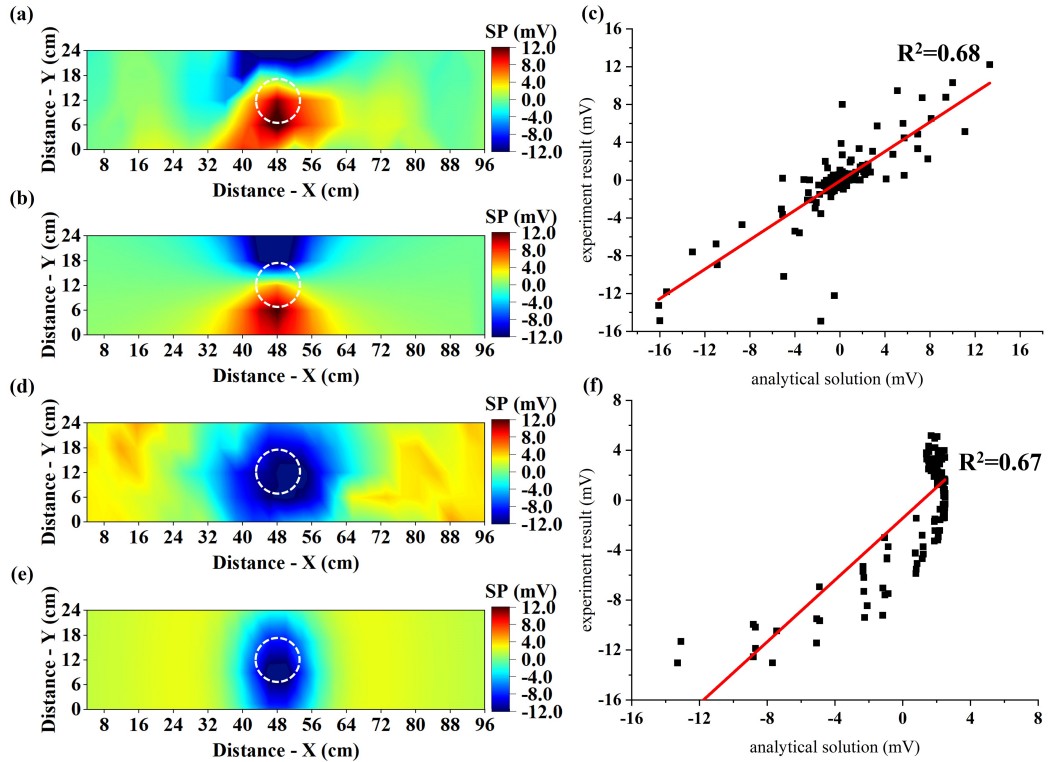

**Figure 8.** Experimental measurements of the self-potential and comparison with forward modeling results. The white dashed circle indicates the projection position of the sphere. (a) Measurement results for the inclined sphere. (b) Forward modeling for the inclined dipole. (c) Comparison of modeling results for the inclined model. (d) Measurement results for the vertically placed sphere. (e) Forward modeling for the vertical dipole. (f) Comparison of modeling results for the vertical model.

## 5 Conclusions

The three-dimensional (3D) self-potential (SP) analytical solution for regularly polarized bodies in a layered seafloor model is pivotal for advancing mineral exploration and enhancing the forward modeling capabilities of the SP method. In this study, we derived a comprehensive 3D analytical solution using the mirror image method. This approach effectively reflects the self-potential signal characteristics of simply polarized bodies in layered media, while also addressing the issue of field survey lines may not being directly above the polarization center. By examining the equivalence between a sphere and an electric dipole, we derived formulas for two-layer and three-layer models by superposing the scalar fields generated by the source and mirror images in different media. The validity of the mirror image method was confirmed through a comparison of the two-layer model's analytical solution with the 2D analytical solution for an uniform space, demonstrating remarkable consistency. We conducted a laboratory experiment simulating a simplified SMS model. By varying the angle of a Fe-Cu sphere interface with respect to the *XOY* plane, we investigated different electric field distributions. The comparison between the measured data and

the 3D analytical solution showed a high degree of agreement. It indicate that the analytical solution based on the mirror image method is highly effective for forward modeling in SMS exploration. This method not only provides a rigorous solution but also ensures faster computational performance compared to iterative numerical methods. Consequently, it offers a solid foundation for the inversion and interpretation of measured SP data, ultimately contributing to more accurate and efficient exploration of seafloor massive sulfide deposits.

*Code availability.* The code are written in Matlab and figures are displayed with Origin software.
The code is available at: https://doi.org/10.5281/zenodo.13913302

*Data availability.* The source data is available at: https://doi.org/10.5281/zenodo.13913302

*Author contributions.* Pengfei Zhang is the first author of this paper. It's part of his research project. He conceived, designed the study, conducted the experiment and wrote the paper. Yi-an Cui designed the study and experiment. Jing Xie and Youjun Guo developed the idea for the study and did some analyses about the data. Jieran Liu collected the experiment data. All authors discussed the results and revised the manuscript.

*Competing interests.* The contact author has declared that none of the authors has any competing interests

*Acknowledgements.* This work was financially supported by the National Natural Science Foundation of China (42174170, 41874145, 72088101).

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
