# Peer review of "Three-Dimensional Analytical Solution of Self-potential from Regularly Polarized Bodies in Layered Seafloor Model"

_Geoscientific Model Development, 2023_

## Referee Comment (RC2)

**OBSERVATIONS**

This paper proposed a method to solve the analytical solution of the SP and based on the mirror image current theory, the 2D and 3D analytical solution formulas were derived. Some things in the paper were not clear and properly stated which made the paper difficult to read especially the derivation of the formulas which is the major goal of the paper. Below are some of my observations;

1. I thought the formula accords with the Laplace equation in spherical coordinates is given by

$$\frac{1}{r^2}\frac{\partial}{\partial r}\left(r^2\frac{\partial u}{\partial r}\right) + \frac{1}{r^2}\frac{1}{sin\theta}\frac{\partial}{\partial \theta}\left(sin\theta\frac{\partial u}{\partial \theta}\right) = 0$$

What happened to the denominators $(r^2)$ in your own quoted equation?

2. Was equation 3 (the general solution of potential) in section 2.1 something you guys came up with, or was it already there? If the equation is already known, you ought to credit the authors or the source, in my opinion. Although equations 1.6 and 1.7 were mentioned, your manuscript does not contain any equations of that kind.

3. What steps led from equation 7 to equation 8? Is it feasible to simplify it and the connectivity for comprehension? If you have derived the majority of these equations, please state so. If not, provide appropriate citation by quoting the academics.

4. For someone who is not in the geosciences, most of the derivations might be unclear for them. I believe you should correctly demonstrate the connections and how you arrived to equation 13. All I'm asking is that you provide a step-by-step explanation of the derivations as this is your new formula, so that other researchers can comprehend the equations and draw conclusions from them. The Table 1 formulas must to be clearly described or demonstrate how you arrived at each of their instances.

5. Is it possible to put the result analysis of Section 3 and Section 4 in table format to support the figure results? The table will help to make the analysis more comprehensible.

6. For better understanding, it would be beneficial if you could explain section 3 following figure 4, and each figure should have an explanation. Section 4 should follow the same procedure as related to Figure 5. Provide a thorough

explanation for each of the following three figures: a horizontal electric dipole, b vertical electric dipole, and c tilted electric dipole.

7. It is not clear if you compared the formula for the 2D analytical solution you generated with the 2D measured data, just as you did for the 3D analytical solution

However, to make it easier, it would be even better if the previously mentioned points—particularly the derivation formula and the results—were broken down and discussed in detail while keeping in mind potential readers of the work who are not geoscientists. The current state of the document may make it somewhat difficult for someone who is not in the field of geoscience to understand, and the goal of any research paper is to provide clarity so that others can benefit from your work.

---

## Author Comment (AC1)

Central South University

No.932, Lushannan Road

Changsha

China

Email: symdwjz@foxmail.com

5$^{th}$ July 2024

Dear editor and reviewer,

Re:" **Three-Dimensional Analytical Solution of Self-potential from Regularly Polarized Bodies in Layered Seafloor Model**"

Thank you for taking time to review our paper in such detail. We appreciate the chance to revise this paper.

Through your review, we have recognized numerous deficiencies in the manuscript. In response to the reviewers' comments, we have made extensive revisions to the manuscript. We have supplemented the content of the formula derivations, adding more detailed steps to make them easier to understand. In Section 2.1, we demonstrated the equivalence of the self-potential generated by a uniformly polarized sphere and that generated by an electric dipole. Subsequently, we derived the three-dimensional analytical solutions of self-potential in two-layer and three-layer media. In Section 3, we revised all the figures, using new three-dimensional potential distribution plots to better present the results of the analytical solutions. We also added substantial explanatory content to improve the readability of the article. In Section 4, we provided a more detailed description of the experimental process. Recognizing the limitations of previously comparing the central survey line results, we have now compared the experimental results with the two-dimensional slices of the analytical solutions and evaluated them by calculating the coefficient of determination($R^2$). Additionally, we corrected erroneous descriptions in the manuscript and standardized the notation, including symbols and Greek letters.

In this response letter, your comments in blue and our response in black. Because we have made extensive revisions, it may not be possible to showcase all the changes through the discussion alone. Therefore, we have attached the revised manuscript.

This paper proposed a method to solve the analytical solution of the SP and based on the mirror image current theory, the 2D and 3D analytical solution formulas were derived. Some things in the paper were not clear and properly stated which made the paper difficult to read especially the derivation of the formulas which is the major goal of the paper. Below are some of my observations

Thank you for your insightful comment regarding the presence of $1/r^2$ terms in the Laplace equation in spherical coordinates. I would like to clarify that the formula in question is indeed derived within the context of spherical coordinates. In a uniformly polarized sphere, the external potential U is distributed symmetrically about the polarization axis and is independent of the azimuthal angle. This specific symmetry leads to a simplified form of the Laplace equation that we have used in our derivations.

he Laplace equation in spherical coordinates $(r,\theta,\phi)$ is given by:

$$\nabla^2 U = \frac{1}{r^2}\frac{\partial}{\partial r}\left(r^2\frac{\partial U}{\partial r}\right)+\frac{1}{r^2\sin\theta}\frac{\partial}{\partial\theta}\left(\sin\theta\frac{\partial U}{\partial\theta}\right)+\frac{1}{r^2\sin^2\theta}\frac{\partial^2 U}{\partial\phi^2}=0$$

Given the problem's symmetry, the potential U is independent of the azimuthal angle $\phi$. Thus, the equation reduces to:

$$\nabla^2 U = \frac{1}{r^2}\frac{\partial}{\partial r}\left(r^2\frac{\partial U}{\partial r}\right)+\frac{1}{r^2\sin\theta}\frac{\partial}{\partial\theta}\left(\sin\theta\frac{\partial U}{\partial\theta}\right)=0$$

In our derivation, we focus on solving the radial part of the equation:

$$\frac{1}{r^2}\frac{\partial}{\partial r}\left(r^2\frac{\partial U}{\partial r}\right)=0$$

This can be rewritten as:

$$\frac{\partial}{\partial r}\left(r^2\frac{\partial U}{\partial r}\right)=0$$

Integrating this equation with respect to $r$:

$$r^2\frac{\partial U}{\partial r}=A$$

where $A$ is an integration constant. Solving for $U$:

$$U(r)=-\frac{A}{r}+B$$

where $B$ is another integration constant.

When we present the final expression for the potential U(r), the term A/r results from the integration process. The $1/r^2$ term is inherent in the integration steps and thus does not explicitly appear in the final potential expression.

This form of the equation aligns with the theoretical framework described in our paper. Similar derivations and explanations can be found in classical works on potential theory, such as those by MacMillan (1958) in "The Theory of the Potential".

I hope this explanation clarifies the basis of our derivations and demonstrates that the formulas presented are consistent with the Laplace equation in spherical coordinates. We will incorporate additional explanations in the revised manuscript to clarify the derivation steps and the context in which the $1/r^2$ terms are inherently accounted for due to the spherical symmetry of the problem.

2.Was equation 3 (the general solution of potential) in section 2.1 something you guys came up with, or was it already there? If the equation is already known, you ought to credit the authors or the source, in my opinion. Although equations 1.6 and 1.7 were mentioned, your manuscript does not contain any equations of that kind.

Equation 3 was derived using the method of separation of variables. Here, we have provided the complete solution process.

The Laplace equation in spherical coordinates is given by:

$$\frac{\partial}{\partial R}\left( R^2 \frac{\partial U}{\partial R} \right) + \frac{1}{\sin\theta} \frac{\partial}{\partial\theta}\left( \sin\theta \frac{\partial U}{\partial\theta} \right) = 0$$

We want to find a solution $U(R,\theta)$ in the form:

$$U(R,\theta) = R^n P_n(\cos\theta)$$

where $P_n(\cos\theta))$ are the Legendre polynomials.

Assume the solution can be separated into a radial part $R(R)$ and an angular part $\Theta(\theta)$:

$$U(R,\theta) = R(R)\Theta(\theta)$$

Substitute this into the Laplace equation:

$$\frac{\partial}{\partial R}\left( R^2 \frac{\partial(R(R)\Theta(\theta))}{\partial R} \right) + \frac{1}{\sin\theta} \frac{\partial}{\partial\theta}\left( \sin\theta \frac{\partial(R(R)\Theta(\theta))}{\partial\theta} \right) = 0$$

Expand the equation:

$$\Theta(\theta)\frac{\partial}{\partial R}\left( R^2 \frac{\partial R(R)}{\partial R} \right) + R(R)\frac{1}{\sin\theta} \frac{\partial}{\partial\theta}\left( \sin\theta \frac{\partial\Theta(\theta)}{\partial\theta} \right) = 0$$

Separate the equation into two independent parts:

$$\frac{1}{R(R)} \frac{1}{R^2} \frac{d}{dR}\left( R^2 \frac{dR(R)}{dR} \right) = -\frac{1}{\Theta(\theta)} \frac{1}{\sin\theta} \frac{d}{d\theta}\left( \sin\theta \frac{d\Theta(\theta)}{d\theta} \right) = \lambda$$

where $\lambda$ is the separation constant.

The angular part of the equation is:

$$\frac{1}{\sin\theta}\frac{d}{d\theta}\left(\sin\theta\frac{d\Theta(\theta)}{d\theta}\right)+\lambda\Theta(\theta)=0$$

The solution to this equation is the Legendre polynomial:

$$\Theta(\theta)=P_n(\cos\theta)$$

where $\lambda=n(n+1)$ and n is an integer.

The radial part of the equation is:

$$\frac{1}{R^2}\frac{d}{dR}\left(R^2\frac{dR(R)}{dR}\right)-\lambda R(R)=0$$

Substitute $\lambda=n(n+1)$:

$$\frac{1}{R^2}\frac{d}{dR}\left(R^2\frac{dR(R)}{dR}\right)-n(n+1)R(R)=0$$

Assume the solution is of the form:

$$R(R)=A_nR^n+\frac{B_n}{R^{n+1}}$$

Combine the radial and angular parts to get the total solution:

$$U(R,\theta)=\sum_{n=0}^{\infty}\left(A_nR^n+\frac{B_n}{R^{n+1}}\right)P_n(\cos\theta)$$

We apologize for our oversight; Equations (1.6) and (1.7) do not exist. It should refer to Equations (9) and (10). We have made the corrections in the main text.

3.What steps led from equation 7 to equation 8? Is it feasible to simplify it and the connectivity for comprehension? If you have derived the majority of these equations, please state so. If not, provide appropriate citation by quoting the academics.

The transition from equation 7 to equation 8 (equation 10 and equation 11 in the revised manuscript) involves the principle of superposition of potentials, which is fundamental in electrostatics and electromagnetic theory. Specifically, the formula provided for $\Phi_{sea}$ represents the combined potential due to a source dipole and its image dipole.

The scalar potential U caused by a constant electric dipole is given by:

$$U=\frac{Idl}{4\pi\sigma}\left(\frac{-x}{R^3}\right)=-P_0\frac{x}{R^3}$$

where $I$ is the current, $dl$ is the length element, $\sigma$ is the conductivity, and R is the distance from the dipole to the observation point.

The potential in a medium with a boundary (such as the sea-air interface) can be determined by considering the contribution from both the real dipole and its image dipole. This method ensures the boundary conditions are satisfied.

The potential in the seawater (z>0) is given by the sum of the potentials due to the real dipole(P=$I_x$d$l$)and the image dipole(P'=$I_x$'d$l$):

$$\Phi_{sea} = \frac{I_x dl(x-x_0)}{4\pi\sigma_1 R_1^3} + \frac{I_x' dl(x-x_0)}{4\pi\sigma_1 R_0^3} \quad (z>0)$$

$R_1$ and $R_0$ are the distances from the observation point to the real and image dipoles.

4.For someone who is not in the geosciences, most of the derivations might be unclear for them. I believe you should correctly demonstrate the connections and how you arrived to equation 13. All I'm asking is that you provide a step-by-step explanation of the derivations as this is your new formula, so that other researchers can comprehend the equations and draw conclusions from them. The Table 1 formulas must to be clearly described or demonstrate how you arrived at each of their instances.

Thanks for your good suggestion. The image method is a mathematical technique used to simplify boundary value problems in electrostatics. It involves replacing the boundary with an equivalent charge distribution (the image charge or dipole) that ensures the boundary conditions are satisfied. In the paper, we first discuss 2-layer models. Whether it is the seawater-air model or the seawater-seafloor model, the essence of the image method is to replace the boundary with an electric dipole or a virtual image dipole, thereby simplifying the "one source with two mediums" problem to a "one medium with one source and one image source" problem. This is done while satisfying the actual boundary conditions. When the measuring points and the source are in the same medium, the potential calculation must take into account the conductivity of the medium. Therefore, the potential caused by a constant electric dipole is the superposition of the potentials generated by the source point and the image point (located in the other medium). When the measuring point and the source point are not in the same medium, to satisfy the boundary conditions:

1)The potential must be continuous across the boundary. This means that the potential just above the boundary in the air must equal the potential just below the boundary in the seawater

2)The normal component of the current density (or electric field) should also be continuous across the boundary. This condition ensures that the boundary behaves correctly according to Maxwell's equations.

the image dipole coincides with the real dipole. The combined dipole moment is used to calculate the potential in the air. Because the dipole moment of the image dipole is unknown, we need to solve it using the boundary conditions.

In the seawater-air model, the image dipole moments for the two different measurement scenarios are solved as shown in Equations (14) and (15). In the seawater-seafloor model, we can use the same method to solve for the dipole moments in different situations by combining the boundary conditions. When the measuring point is located in seawater:

$$I_x dl''' = \frac{\sigma_1 - \sigma_2}{\sigma_1 + \sigma_2} I_x dl$$

And when the measuring point is located in seafloor,

$$Ixdl'''' = \frac{2\sigma_2}{\sigma_1 + \sigma_2} I_x dl$$

The following diagram is a schematic representation of the two-layer medium after processing using the image method.

[Figure]

**Figure 1.** Schematic representation of the two-layer medium after processing using the image method.

In the three-layer model, the image method generates an infinite number of images. As an example, we consider the first three images (as shown in Figure 2). The source point generates corresponding images in the other two media, referred to as the first image. The generated images in turn create new images in the other medium. For instance, an image dipole 1 generated in the air by the source point will produce a second image dipole 1-1 in the seafloor medium; similarly, an image dipole 2 generated in the seafloor medium will produce another second image dipole 2-2 in the air. This process continues, generating an infinite number of image dipoles. We calculate the potential generated by each image point in a manner similar to the two-layer model, based on the boundary conditions. Upon solving for different image points, we find that the image points can be categorized into four types, based on the depth patterns of the image dipoles. The coordinates of the first type of image dipole are $(x_0, y_0, 2mD - z_0)$, $m = 1, 2, \ldots$ , with the corresponding dipole moment solved as

$\left( \dfrac{\sigma_1 - \sigma_2}{\sigma_1 + \sigma_2} \right)^m I_x dl = \eta^m I_x dli, m = 1, 2, \ldots$ . The vector distance between the measuring point and

each image point can be expressed as: $\mathbf{r}_{1m} = (x - x_0)\mathbf{i} + (y - y_0)\mathbf{j} + (z - 2mD + z_0)\mathbf{k}$. The image 2 and image 2-2-2 in the figure represents this type of image. The second type of image dipole located at $(x_0, y_0, 2mD + z_0), m = 1, 2, ...$, with the corresponding dipole moment solved as

$$\left(\frac{\sigma_1 - \sigma_2}{\sigma_1 + \sigma_2}\right)^m I_x dl = \eta^m I_x dli, m = 1, 2, .... $$ The vector distance between the measuring point and

each image point can be expressed as: $\mathbf{r}_{2m} = (x - x_0)\mathbf{i} + (y - y_0)\mathbf{j} + (z - 2mD - z_0)\mathbf{k}$. The image 1-1 represents this type. The third type of image dipole located at $(x_0, y_0, -2nD + z_0), n = 0, 1, ...$, with the corresponding dipole moment

$$\left(\frac{\sigma_1 - \sigma_2}{\sigma_1 + \sigma_2}\right)^n I_x dl = \eta^n I_x dli, n = 0, 1, .... $$ The vector distance between the measuring point and

each image point can be expressed as: $\mathbf{r}_{1n} = (x - x_0)\mathbf{i} + (y - y_0)\mathbf{j} + (z + 2nD - z_0)\mathbf{k}$. The source and image2-2 represent this type. The coordinates of the first type of image dipole are $(x_0, y_0, -2nD - z_0), n = 0, 1, ...$, with the corresponding dipole moment

$$\left(\frac{\sigma_1 - \sigma_2}{\sigma_1 + \sigma_2}\right)^n I_x dl = \eta^n I_x dli, n = 0, 1, .... $$ The vector distance between the measuring point and

each image point can be expressed as: $\mathbf{r}_{2n} = (x - x_0)\mathbf{i} + (y - y_0)\mathbf{j} + (z + 2nD + z_0)\mathbf{k}$. The image 1 and image 1-1-1 represent this type of image. The classification of these image points is based on their positions. We can see that, apart from the source point, the third and fourth types of image points are located in the air, while the first and second types are located in the seafloor. Other than this, there are no other physical differences between the various types of image points.

[Figure]

**Figure 2**. Schematic representation of the three-layer medium after processing using the image method (Using the first three image points as an example).

5.Is it possible to put the result analysis of Section 3 and Section 4 in table format to support the figure results? The table will help to make the analysis more comprehensible.

We understand the importance of presenting results in a clear and comprehensible manner. In our revised manuscript, we have provided detailed analyses in Section 3 and Section 4 using three-dimensional potential distributions and depth slices, as well as calculating the $R^2$ values in section 4. These visual representations are crucial for illustrating the spatial variations and correlations inherent in our study. We believe that the three-dimensional figures and depth slices offer a more intuitive understanding of the spatial distribution of self-potential signals, which might be challenging to convey effectively in a tabular format. Additionally, the $R^2$ values calculated and presented in Section 4 quantitatively support the accuracy and correlation of our model predictions with the experimental data. We hope this explanation clarifies our approach and the rationale behind our presentation choices.

6.For better understanding, it would be beneficial if you could explain section 3 following figure 4, and each figure should have an explanation. Section 4 should follow the same procedure as related to Figure 5. Provide a thorough explanation for each of the following three figures: a horizontal electric dipole, b vertical electric dipole, and c tilted electric dipole.

In Section 3, we have supplemented extensive explanatory content, including the interpretation of self-potential distributions generated by horizontal, vertical, and inclined dipoles. We have conducted a detailed analysis of self-potential slices at different depths. In Section 4, we have added descriptions of the experimental setup and procedures, and we have compared and evaluated the measured self-potential results with the forward modeling results.

7.It is not clear if you compared the formula for the 2D analytical solution you generated with the 2D measured data, just as you did for the 3D analytical solution.

We thank you for your insightful suggestion. The primary focus of our research is on three-dimensional (3D) analysis, which is why we developed and presented the 3D analytical solution. Correspondingly, we conducted 3D physical simulations to validate our analytical models. Our experimental setup is inherently three-dimensional, with the entire experiment conducted in a 3D space where self-potential can be recorded at any point within this space.

We realize that our initial description might have been unclear and potentially confusing. To address this, we have revised the relevant sections of the manuscript to better emphasize our 3D approach and its importance to our study.

The revisions clarify that the core objective was to establish and validate a comprehensive 3D analytical solution. The physical simulations were also designed in a 3D context to provide accurate validation of these models.

We hope this detailed explanation clarifies our approach and addresses your concern. Thank you for your constructive feedback, which has been instrumental in refining our study.

However, to make it easier, it would be even better if the previously mentioned points—particularly the derivation formula and the results—were broken down and discussed in detail while keeping in mind potential readers of the work who are not geoscientists. The current state of the document may make it somewhat difficult for someone who is not in the field of geoscience to understand, and the goal of any research paper is to provide clarity so that others can benefit from your work.

---

## Author Comment (AC2)

Central South University

No.932, Lushannan Road

Changsha

China

Email: symdwjz@foxmail.com

4$^{th}$ July 2024

Dear editor and reviewer,

Re:" **Three-Dimensional Analytical Solution of Self-potential from Regularly Polarized Bodies in Layered Seafloor Model**"

Thank you for taking time to review our paper in such detail. We appreciate the chance to revise this paper.

Through your review, we have identified several deficiencies in the manuscript. In response to the reviewers' comments, we have made extensive revisions. We have supplemented the content of the formula derivations, adding more detailed steps to enhance understanding. And we revised all the figures, incorporating new three-dimensional potential distribution plots to more effectively present the analytical solution results. We also added substantial explanatory content to improve the article's readability. In Section 4, we provided a more detailed description of the experimental process.

In this response letter, your comments are highlighted in blue, and our responses are in black.

The authors introduced a 3D analytical solution for forwarding modeling of self-potential from sulfide deposits. Although the topic is interesting and the overall flow of the paper, from the formulation to comparison with 2D analytical solution and experimental results, is great, I can hardly say the paper is well written in general and there seems to be a lot of room for improvement of the manuscript.

The motivation of the current model development is not compelling, especially given that (1) the system has not been proven to be applicable to the field (even though the authors attempted to validate the model with laboratory experiments, it does not necessarily mean it can be used in the field) and (2) complicated situations are likely to be more easily dealt with numerical tools/methods and there seem to be some published already.

Formulation is poorly explained especially given that deriving analytical solution for the system should be the essential part for a model development paper.    I found it difficult to follow how the equations are derived. There can be several reasons for this, including (1) the papers cited (Li et al., 2005, He, 2012) for seemingly essential equations are not publicly available and not

explained well, (2) the authors do not seem to use mathematical symbols in an organized way, including seemingly interchangeable and thus confusing use of uppercase/lowercase and/or Greek/English letters and/or vector/scalar expressions, and (3) figure to help the reader to understand formulation is lacking and/or not referenced.

Especially, please consider making clearer explanations on (1) what is proved and how this is done in Section 2.1, (2) the mirror image method in general potentially with a specific diagram/schematic in Section 2.2 and (3) derivation of one of formulation from Table 1 in Section 2.3.

Thanks for your suggestions.

Laboratory settings allow for controlled conditions where variables can be systematically tested and understood, providing a strong foundation for subsequent field tests. We plan to extend our research to field applications in future studies, following this step-wise approach. This methodology aligns with established scientific practices where theoretical models are first validated in controlled environments before being tested in more complex, real-world scenarios. Studies have shown that theoretical models validated in laboratory settings can be successfully adapted for field applications. For instance, Ishido and Mizutani (1981) validated SP methods using laboratory simulations before successful field deployment (Ishido, T., & Mizutani, H. (1981). Experimental and theoretical basis of electrokinetic phenomena in rock-water systems and its applications to geophysics. Journal of Geophysical Research: Solid Earth, 86(B3), 1763-1775). Andre et al. developed a sandbox experiment to monitor the evolution of a self-potential anomaly associated with redox processes. These precedents provide a strong basis for our approach (Revil, A., Su, Z., Zhu, Z., & Maineult, A. (2023). Self-Potential as a Tool to Monitor Redox Reactions at an Ore Body: A Sandbox Experiment. *Minerals*, *13*(6), 716.).

While numerical methods are indeed powerful and versatile, they also have limitations. Numerical solutions often involve approximations and are influenced by boundary conditions and source configurations. For instance, Stoll and Bryan (1970) highlighted the complexities in solving stiffness matrices and the limitations posed by artificial boundary conditions in numerical methods (Stoll, R. D., & Bryan, G. M. (1970). Wave attenuation in saturated sediments. The Journal of the Acoustical Society of America, 47(5B), 1440-1447). Our model, based on the mirror image method, offers a strict analytical solution that avoids these numerical approximations, providing exact results under defined conditions.

Therefore, analytical solutions provide exact results, which are critical for validating numerical models and ensuring their accuracy. They can serve as benchmarks for numerical simulations, enabling us to identify and correct deviations in numerical approaches. In scenarios where the analytical model is applicable, it offers faster computations compared to iterative numerical methods.

For the derivation process, we have supplemented with a more comprehensive derivation and

final formula. We have corrected the citations for the referenced papers. Equation (7) (equation (10) in revised manuscript), which represents the formula for the scalar potential caused by a constant electric dipole, can be supported by several academic sources (For example, here's a detailed derivation in UTA

*https://web2.ph.utexas.edu/~vadim/Classes/2017f/dipole.pdf#:~:text=URL%3A%20https%3A %2F%2Fweb2.ph.utexas.edu%2F~vadim%2FClasses%2F2017f%2Fdipole.pdf%0AVisible% 3A%200%25%20*). This should provide a more solid theoretical foundation and address any concerns about the validity of the equation. We also recognize that providing a reference to an unpublished document for an important formula is inappropriate. Considering that this is a classical formula for an electric dipole in electromagnetic physics, we believe it is more appropriate not to include a reference here.

As indicated by the title of Section 2.1, this section demonstrates that the self-potential generated by a regularly polarized sphere can be equivalently represented by the self-potential generated by an electric dipole. Through this demonstration, we can directly use the self-potential formula for an equivalent electric dipole when calculating the self-potential of regularly polarized bodies.

We have made substantial revisions and additions to the derivation process and descriptions in order to clarify our derivation process. Here, we provide a supplementary explanation of the entire derivation process. When there are two media, a source point will generate another image point on the opposite side of the interface between the two media. We derived the potential of this image point using boundary conditions. In the case of three layers of media, the source point generates image points on the opposite sides of both interfaces, and these image points will, in turn, generate new image points in the other media. For example, a source point in a seawater-air combination generates an image point (image point 1) in the air, which subsequently generates another image point (image point 1-1) in the seafloor medium. Similarly, a source point in a seawater-seafloor combination generates an image point (image point 2) in the seafloor, and this image point generates another image point (image point 2-1) in the air. This process continues iteratively. During the derivation, we found that the self-potential expressions for different image points can be categorized into four types. These four types of dipoles are distinguished by different dipole moments (as shown in Table 1 of the main text). These image points themselves do not have physical significance but are mathematically differentiated.

We acknowledge the confusion caused by the inconsistent use of mathematical symbols. We have revised the manuscript to use mathematical symbols in an organized and consistent manner, distinguishing clearly between uppercase/lowercase, Greek/English letters, and vector/scalar expressions. This should enhance readability and reduce confusion.

L9. "It is significant to fast and precise forward modeling and inversion for SMS explorations.".
I think English grammar here is not correct.

We have revised it to: "This approach is significant for achieving fast and precise forward modeling and inversion in SMS explorations."

L22. What is "AUV"?

AUV is an abbreviation for autonomous underwater vehicle. AUV is equipped with advanced navigation and positioning systems, allowing for precise localization in complex seafloor environments. By integrating various sensing devices, such as natural potential instruments and magnetometers, they can conduct comprehensive surveys and acquire diverse sets of data.

We have recognized that using abbreviations here is inappropriate, and we have made corrections in the main text.

L34. What do you mean by "affection of the field source"?

Our description here is unclear. In the numerical forward modeling of the self-potential method, using the finite element method as an example:

$$Kx = f$$

where K is the stiffness matrix, x is the potential distribution, and f is the source term.

The source points (such as polarization bodies or charge distributions) indeed affect the overall numerical solution process, but this influence primarily manifests in the right-hand side term (source term) of the system equation, rather than directly altering the stiffness matrix. The stiffness matrix is mainly determined by the physical properties and structure of the medium. For an anomalous body with uneven surface polarization intensity, the right-hand side term (source term) will change due to this uncertainty. This is one of the limitations of numerical forward modeling.

We have revised this section in the main text.

L34. "which occur difficulty in solving the Poisson equation." I think the authors should correct English.

We have revised it to: "Compared to numerical methods, analytical solutions are strict formulas that can overcome the difficulties in solving the Poisson equation."

L58. What is the definition of H?

We apologize for the oversight; this should be the seawater depth (D) instead of H.

L79. rho_0 must be a typo of r_0?

"We apologize for our writing error; it should be $r_0$ at this point. We have corrected this in the main text.

L82. "So we get the potential distribution along the surface of a uniformly polarized sphere is equivalent to an electric dipole." I do not understand what this means.

Through the derivation of the above formulas, we conclude that under the conditions of R=r and $P_0$=M, the self-potential distribution of a uniformly polarized spherical body is the same as that of a dipole. For clarity, we have made modifications in the main text:" Therefore, we can conclude that the potential distribution of a uniformly polarized spherical body is the same as that of a dipole."

L92. Is the use of permittivity correct here? In other words, do permittivity and conductivity have the same units?

Use of $\sigma_1$ in the Ocean:

In the ocean, the medium is conductive. The conductivity of seawater is typically denoted by $\sigma_1$. This is appropriate because the electric field behavior in a conductive medium is influenced by its conductivity. The formula (11) involving $\sigma_1$ is used to describe the scalar potential and electric field distribution in seawater, which aligns with the medium's conductive nature.

Use of $\varepsilon_0$ in the Air:

In the air, the medium is not conductive but rather dielectric. The permittivity of free space is denoted by $\varepsilon_0$. When the electric field extends into the air, the potential expression uses $\varepsilon_0$ because the permittivity influences the behavior of electric fields in a dielectric medium.

Given this context, the use of $\sigma_1$ for the ocean and $\varepsilon_0$ for the air is intentional and correct. It reflects the different physical properties of the two media — conductivity for seawater and permittivity for air.

And permittivity and conductivity do not have the same units. The unit of permittivity is farads per meter (F/m) and the unit of conductivity is siemens per meter (S/m). In our paper, distinguishing between these two physical quantities is to illustrate the uniqueness of solving the natural potential in a two-layer air-seawater model.

L137. What do you mean by "the dipole moment is 1 D."?

In some papers, the electric dipole moment is expressed in Debye (D). We have recognized that using D here is inappropriate, and we have revised it to the more standard unit of Coulomb meters (C·m).

L138. 10-10 must mean 10^-10.   What are units and adjustment term exactly?

Your understanding is correct; it should be $10^{-10}$ here. This oversight was due to a formatting error in the LaTeX typesetting of our paper. We will provide further clarification on this statement.

In the formulas for the three-layer model, the potential calculation is an iterative process. We set the iteration condition such that the iteration stops when the difference between the potential values of successive calculations is less than $10^{-10}$ millivolts. In practice, $10^{-10}$ is a very small value that adequately ensures both computational efficiency and accuracy.

L140. Do "exceptions" mean negative/positive peaks?

Your understanding is correct; this refers to a horizontal electric dipole producing potential

signals of equal magnitude but opposite sign. We have revised the expression in the main text: The magnitude of the absolute values is equal.

Our previous expression was inaccurate. An inclined polarized electric dipole generates self-potential signals of different absolute values on either side of the dipole. The self-potential signals increase along the polarization angle of the dipole.

L147. The authors should describe exactly what redox reaction is occurring.

In the main text, we have supplemented the specific description of the redox reaction. The prevailing explanation for the origin of the self-potential in seafloor hydrothermal sulfide deposits is the geobattery model formed by the redox reactions of sulfide minerals. We simulated this redox reaction using an iron-copper composite sphere. Here, we have added a graph showing the variation of the self-potential signals of 10 electrodes placed directly above the inclined sphere. In this graph, it can be observed that the self-potential polarities on either side of the sphere's polarization direction are opposite, with similar trends in their variation. We selected the self-potential results at 5000 seconds to plot the self-potential slices in Figure 8 of the main text. At this time, the self-potential is relatively stable.

[Figure]

**Figure 1.** Temporal self-potential curves for electrodes located above the sphere.

[Figure]

**Figure 2.** Sketch of the experimental observation electrodes, with the red dashed line

indicating the projection position of the sphere. Electrodes 60 and 65 are marked.

---

## Author Response (AR2)

Central South University
No.932, Lushannan Road
Changsha
China

Email: symdwjz@foxmail.com

11[th] October 2024

Dear editor,

Re: "**Three-Dimensional Analytical Solution of Self-potential from Regularly Polarized Bodies in Layered Seafloor Model**"

Thank you very much for your valuable comments and suggestions regarding our paper. We have supplemented the literature review section, adding more relevant studies to provide a more comprehensive background and theoretical foundation for the paper. And we have added Table 2 to present the experimental results, including statistical information such as the mean and standard deviation. This addition ensures that the results are more clearly and comprehensively presented. The code and data have been published on Zenodo, and we have included the corresponding DOI link in the paper to allow readers to access and reproduce our results.

Thank you again for your review and constructive feedback. We hope these revisions meet your expectations and look forward to any further feedback you may have.

Best regards,

Pengfei Zhang